# ADVERSARIALLY LEARNED ANOMALY DETECTION FOR TIME SERIES DATA

## ABSTRACT

Anomaly detection in time series data is an important topic in many domains. However, time series are known to be particularly hard to analyze. Based on the recent developments in adversarially learned models, we propose a new approach for anomaly detection in time series data. We build upon the idea to use a combination of a reconstruction error and the output of a *Critic* network. To this end we propose a cycle-consistent GAN architecture for sequential data and a new way of measuring the reconstruction error. We then show in a detailed evaluation how the different parts of our model contribute to the final anomaly score and demonstrate how the method improves the results on several data sets. We also compare our model to other baseline anomaly detection methods to verify its performance.

## 1 INTRODUCTION

With recent proliferation of devices collecting temporal observations, there has been an increasing demand for anomaly detection in time series. The main goal is to identify any behavior in the time series that is unusual, flag it and bring it for analysis. In many real world settings a continuous time series collected over long periods is provided, and the goal is to isolate *anomalous* sub sequences of varied lengths. One usually does not know where those sub sequences may exist, or how long or short each one would be and how many there are.

In a classical setting, it is possible to segment the time series into many subsequences (overlapping or otherwise) of a certain length and apply methods focused on generating an anomaly score for each sequence in order to show how certain sequences are compared to others. Chandola et al. (2008) present a comparative study for several time series anomaly detection methods. They categorize the techniques into *kernel*-based, *window*-based and *Markovian* techniques.

Practitioners who don't know how to segment may resort to less complex techniques, such as simple thresholding, to detect any data points that exceed the normal range (Chandola et al., 2009). However, many anomalies do not exceed any boundaries – for example, they may have values that are purportedly "normal," but are actually unusual at the specific time that they occur. These contextual anomalies are naturally harder to identify, since the context of a signal is often not clear (Chandola et al., 2009; Ahmad et al., 2017).

In recent years, Deep Learning-based methods have been developed to deal with such issues (Kwon et al., 2017). These methods make use of the increased availability of data in order to learn the underlying structure of a time series, and to identify unusual changes in behavior using prediction or reconstruction errors (Malhotra et al., 2015). Within this framework – learning a model, predicting and reconstructing sequences, and using reconstruction errors to detect anomalies – multiple variants have been developed (Malhotra et al., 2015; Hundman et al., 2018; Goh et al., 2017).

At the same time, recent years have also seen the introduction of adversarially trained networks, which can learn the underlying distributions of data sets and generate impressive synthetic data from this information. Generative Adversarial Networks (GANs), which were introduced by Goodfellow et al. (2014) in 2014, have been very successful, especially in the area of image processing. Without direct access to real data, *Generators* in GANs attempt to synthesize real-*looking* data by implicitly learning the structure of a dataset. Seeing this success has motivated us to explore whether GANs can also learn the structure of a time series. To the best of our knowledge, only one other work by Li et al. (2018) uses GANs in time series anomaly detection. Building on this approach, this paper

aims to give more thorough insight into this domain, and to demonstrate how adversarially learned networks could be used for anomaly detection in time series data.

Our key contribution is the development of a cycle-consistent GAN for sequential data that can be used for anomaly detection. Because we analyze time series data, which naturally comes with short- or long-term dependencies, our encoding and generating networks are based on Long Short Term Memory (LSTM) cells. In order to achieve cycle consistency during training, we use a reconstruction loss for the Encoder and Generator training, and a second *Critic* network to support the correct bidirectional mappings.

Furthermore, we propose that the point-wise reconstruction error between original time series points and the reconstructed points, which is often used for anomaly detection, regularly fails to give the best error function for time series data. Instead, we introduce two similarity measures, which try to evaluate the local similarity between the original and the reconstructed sequences. We then combine this similarity measure and the *Critic* output into a function that gives robust anomaly scores for the time series.

To provide further insights into anomaly detection with GANs and to demonstrate our proposed model, we provide an evaluation which investigates how each component of our model contributes to anomaly detection performance. Finally, we provide several benchmarks on well-known time series data sets and show how our approach exceeds the performance of current state-of-the-art methods.

The paper is structured as follows: First, we give an overview of related literature in section 2. Next, we introduce our model in section 3. We then describe the anomaly detection method in section 4, and give a evaluation of our proposed model in section 5.

## 2  RELATED WORK

Anomaly detection is a broad area of study, and several methods have been developed over the course of years, if not decades. To limit the scope of related work, we focus on instances where generative adversarial networks have been used for anomaly detection.

GANs are most often used to generate images and a few studies have used them to detect anomalies in them. For example, Schlegl et al. (2019) and Deecke et al. (2018) use GAN based architectures to detect anomalies in images.

However, only a few studies have used GANs for time series data. For example, Esteban et al. (2017) use recurrent conditional GANs to generate medical time series data. More recently, (Luo et al., 2018) used them to impute missing values in multivariate time series.

In order to use GANs for anomaly detection in time series, Li et al. (2018) proposed a GAN model to capture the distribution of a multivariate time series. The *Critic* is then used to detect anomalies. They also attempt to use the reconstruction loss as an additional anomaly detection method, so they find the inverse mapping from the data space to the latent space. This mapping, which tries to infer $z$ from $X$, is done in a second step after the GAN is trained.

The same two-step approach is used in a more recent preprint by Li et al. (2019) as well as by Schlegl et al. (2017) to detect anomalies in medical images. However, as Zenati et al. (2018) mention in their paper, this method is impractical for large data sets or real-time applications. They propose a bi-directional GAN for anomaly detection in tabular and image data sets, which allows to train the inverse mapping through an encoding network simultaneously.

The idea of training both encoder and decoder networks was originally developed by Donahue et al. (2017) and Dumoulin et al. (2017), who show how to get bidirectional GANs by trying to match joint distributions. In the optimum situation, the joint distributions are the same and the Encoder and Decoder must be inverses of each other. A similar cycle consistent GAN was introduced in Zhu et al. (2017), where two networks try to map into opposite dimensions, such that samples can be mapped from one space to the other and vice versa. Also, Zhou et al. (2019) are using a Encoder and Decoder GAN architecture that allows to use the reconstruction error for anomaly detection in ECG data.

## 3 ADVERSARIAL LEARNING FOR TIME SERIES

While our time series anomaly detection model builds heavily on the bidirectional concepts introduced in the works mentioned above, we are introducing some changes to the loss and objective functions. Therefore we will describe the details of our model in this section.

The standard GAN can be used to generate time series. It consists of two components, the *Generator* $\mathcal{G}$ and the *Critic* (or *Discriminator*) $\mathcal{C}_x$. Typically, both are implemented through neural networks. $\mathcal{G}$ maps $z$ drawn from an assumed latent distribution $\mathbb{P}_Z$ - typically a standard multivariate normal distribution, i. e. $z \sim \mathbb{P}_Z = \mathcal{N}(0, I)$. The input data domain $X$ is described by the given training samples $\{(x_i^{1...t})\}_{i=1}^N$, $x_i^{1...t} \in X$ and $X$ represents the possible time series sequences of length $t$. For convenience of notation we use $x_i$ to imply a time sequence of length $t$.

The goal of $\mathcal{C}_x$ is to distinguish between real time series sequences from $X$ and the generated time series sequences from $\mathcal{G}(z)$, whereas $\mathcal{G}$ is trying to fool $\mathcal{C}_x$ by generating real-looking sequences. Therefore, we have a saddle-point problem, where $\mathcal{G}$ and $\mathcal{C}_x$ are competing against each other.

In our implementation we use a Wasserstein loss, first introduced by Arjovsky et al. (2017), which makes use of the Wasserstein-1 distance when training the Critic network. Formally, let $\mathbb{P}_X$ be the distribution over $X$, then we have the following problem:

$$\min_{\mathcal{G}} \max_{\mathcal{C}_x \in \mathbf{C_x}} V_X(\mathcal{C}_x, \mathcal{G}) \tag{1}$$

with

$$V_X(\mathcal{C}_x, \mathcal{G}) = \mathbb{E}_{x \sim \mathbb{P}_X}[\mathcal{C}_x(x)] - \mathbb{E}_{z \sim \mathbb{P}_Z}[\mathcal{C}_x(\mathcal{G}(z)))] \tag{2}$$

where $\mathbf{C_x}$ denotes the set of 1-Lipschitz functions. We use a LSTM based neural network for $\mathcal{G}$ and a 1-D convolutional neural network for $\mathcal{C}_x$ in our implementation.

In order to match samples with representations in the latent space, we train an encoding network $\mathcal{E}$ for the mapping $\mathcal{E} : X \to \mathbb{P}_Z$. Therefore, we compute another loss term with the intention to minimize the L2 norm of the difference between the original and the reconstructed samples:

$$V_{L2}(\mathcal{E}, \mathcal{G}) = \mathbb{E}_{x \sim \mathbb{P}_X}[\|x - \mathcal{G}(\mathcal{E}(x))\|_2] \tag{3}$$

This gives the following objective we try to solve:

$$\min_{\mathcal{G}, \mathcal{E}} \max_{\mathcal{C}_x \in \mathbf{C_x}} V_X(\mathcal{C}_x, \mathcal{G}) + V_{L2}(\mathcal{E}, \mathcal{G}) \tag{4}$$

In this objective function, the *Critic* for the *Generator* is optimized by approximating the Wasserstein-1 distance, and the *Generator* and the *Encoder* are optimized to minimize this distance and the additional L2 objective function.

To support the correct mapping into the latent distribution $\mathbb{P}_Z$, we add a second *Critic* $\mathcal{C}_z$. The resulting architecture becomes similar to the one due to the CycleGAN proposal of Zhu et al. (2017), although with a different loss. Its purpose is to try to distinguish between random latent samples $z \sim \mathbb{P}_Z$ and encoded samples $\mathcal{E}(x)$ with $x \sim \mathbb{P}_X$. Thus, we have the following additional objective using the Wasserstein-1 loss again:

$$\min_{\mathcal{E}} \max_{\mathcal{C}_z \in \mathbf{C_z}} V_Z(\mathcal{C}_z, \mathcal{E}) \tag{5}$$

with

$$V_Z(\mathcal{C}_z, \mathcal{E}) = \mathbb{E}_{z \sim \mathbb{P}_Z}[\mathcal{C}_z(z)] - \mathbb{E}_{x \sim \mathbb{P}_X}[\mathcal{C}_z(\mathcal{E}(x)))] \tag{6}$$

Combining all of the objectives given in (2), (3) and (6) leads to the final MinMax problem:

$$\min_{\{\mathcal{E}, \mathcal{G}\}} \max_{\{\mathcal{C}_x \in \mathbf{C_x}, \mathcal{C}_z \in \mathbf{C_z}\}} V_X(\mathcal{C}_x, \mathcal{G}) + V_{L2}(\mathcal{E}, \mathcal{G}) + V_Z(\mathcal{C}_z, \mathcal{E}) \tag{7}$$

In order to enforce the 1-Lipschitz constraint during training, we apply a gradient penalty regularization term as introduced in Gulrajani et al. (2017), which penalizes gradients not equal to 1.

The full architecture of our model can be seen in Figure 1.

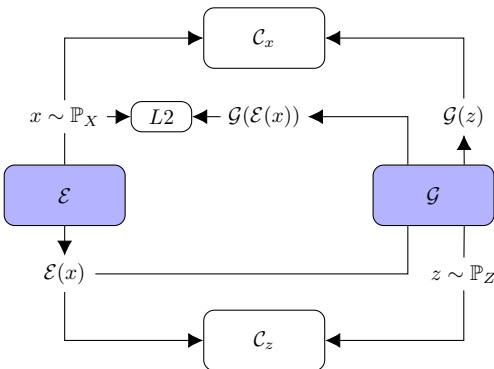

Figure 1: Model architecture

# 4 ANOMALY DETECTION

Having the cycle-consistent architecture, we can now use two different methods in order to detect anomalies. On the one side we can use the reconstruction error by encoding and decoding time series sequences. On the other side we can use the output of the *Critic* $\mathcal{C}_x(x)$ as a direct anomaly measure. In this section, we describe these two methods in more detail and then present a way to combine them effectively to give a robust anomaly score.

## 4.1 RECONSTRUCTION ERROR

Encoding a time series sequence into the latent space using the Encoder and decoding it back into the original data space allows to reconstruct the sequence due to the cycle consistency in our architecture. This part behaves like an Autoencoder, where the reconstruction error can be used as an anomaly measure. The reason to use the reconstruction error is the fact that the model should not be able to reconstruct anomalous sequences as well as normal sequences and the use of this method is well studied and accepted in the area of anomaly detection (Pimentel et al., 2014; Hundman et al., 2018; Malhotra et al., 2015). Typically, the reconstruction error used in anomaly detection for time series is defined as the point-wise difference between the true value and the reconstructed value, i.e. $e_i = x_i - \hat{x}_i, \forall x_i \in X$. However, we claim that this approach might not always be the best way to define the reconstruction error. Instead, we propose a different method, which is applicable in more cases. Informally, we want to measure the local similarity between the true sequence and the reconstructed sequence and identify regions where the similarity is low. There exist various similarity measures for time series (Serrà & Arcos, 2014). One simple reason why we might not want to use the absolute difference between the points is that two curves could have only a small difference but over a long period of time, which would not give high point-wise errors. Thus, we propose two other similarity measures that could be used, i.e. Dynamic Time Warping and the simple difference of areas below the two curves, both applied over windows of certain length in order to measure the similarity locally.

### 4.1.1 DYNAMIC TIME WARPING

Dynamic Time Warping (DTW) was first introduced by Berndt & Clifford (1994). Let us assume that we have two time series $X = (x_1, x_2, \ldots, x_n)$ and $Y = (y_1, y_2, \ldots, y_m)$, and let $M \in \mathbf{R}^{n \times m}$ be a matrix such that the $(i, j)$-th element is a distance measure between $x_i$ and $x_j$, denoted as $w_k$. Then we can construct a warp path $W = (w_1, w_2, \ldots, w_K)$ with $\max(n, m) \leq K < n + m - 1$, and $K$ is the length of the warp path. We want to find the warp path $W^*$ that defines the minimum distance between the two curves, subject to boundary conditions at the start and end, as well as constraints on continuity and monotonicity. The DTW distance between time series $X$ and $Y$ is defined as:

$$W^* = \text{DTW}(X, Y) = \min_W \left[ \frac{1}{K} \sqrt{\sum_{k=1}^{K} w_k} \right] \tag{8}$$

### 4.1.2 AREA DIFFERENCE

For the second, even simpler similarity measure, we want to define the similarity of the two curves by simply comparing the areas beneath them. This seems very intuitive yet not often used in this context, but we will show in our experiments that this approach works very well. Suppose we have two functions $f(x)$ and $g(x)$, then we define the similarity $s$ over an interval $[t, t + l]$ as the average difference of areas below the curves:

$$s_t = \frac{1}{l} \left| \int_t^{t+l} f(x) - g(x) \, dx \right| \tag{9}$$

where $l$ is the length of the sequence that we want to integrate over.

We stress that there is not the absolute difference $|f(x) - g(x)|$ inside of the integral, which would denote the area between the two curves. Instead we allow the integral to become negative, thus allowing the areas between the curves to cancel each other out if the curves cross at some point. By doing so we penalize regions where the original and the predicted sequence are just shifted by a small number.

Since we are only given fixed samples of the functions, we can use the trapezoidal rule to calculate the definite integral. We apply the similarity measure to a moving window of size $n$ (between 50 and 150 worked best in our experiments), which slides over the whole time series. The result is a function of similarity at every point in time, which we can then use as an anomaly measure.

### 4.2 *Critic* OUTPUT

Next to the reconstruction error, we can use the trained Critic $C_x$. During the training process, the *Critic* has to distinguish between real input sequences and synthetic ones. As we use the Wasserstein-1 distance when training the *Critic*, where the *Critic* is intuitively trying to assign scores of realness to the samples, the output of the *Critic* can be seen as a score of how real or fake a sequence is. Therefore, once the *Critic* is trained, it can serve as an anomaly measure for time series sequences.

We show in our experiments that it is indeed the case that the *Critic* assigns different scores to anomalous regions compared to normal regions. This allows to use thresholding techniques to identify the anomalous regions.

### 4.3 ANOMALY SCORE

In order to get a robust anomaly detection measure, we can combine the reconstruction error and the *Critic* output. Different methods have been proposed to combine these two scores. For example, the reconstruction error $RE(x)$ and the *Critic* output $C_x(x)$ can be merged in a single value $a(x)$ with a convex combination, $a(x) = \alpha RE(x) + (1 - \alpha)C_x(x)$ (Li et al., 2018; Schlegl et al., 2017). In our case, we similarly aim to find deviations from the normal state by finding outliers in both scores. To this end, we compute the mean and standard deviation of the two scores and calculate the respective z-score in order to normalize both. Finally, we then multiply both score vectors element-wise:

$$\boldsymbol{a}(x) = RE(x) \odot C_x(x) \tag{10}$$

This way we get very robust anomaly score since high values in both scores are emphasized through the multiplication. Once the score is calculated, thresholding techniques can be applied in order to identify anomalous sequences. In our case, the threshold is calculated over a window of certain length (where we use a window size of 1000 time steps in our experiments). While there exist different approaches to define the threshold, we use a simple static threshold defined as 4 standard deviations from the mean of the window. In order to avoid outliers in both directions during the mean calculation, we define the mean only over the values within the 25th and 75th percentile.

## 5 EXPERIMENTAL RESULTS

### 5.1 DATA SETS

In order to measure the performance of our model, we evaluate it on multiple time series data sets. In total, we have six data sets from two different repositories, all of which contain anomalies.

**Spacecraft telemetry:** Within the spacecraft domain, we have two data sets provided by NASA[1]. The MSL data set contains 27 signals from the Mars Science Laboratory (with a total of 36 anomalies), while the SMAP data set contains 55 signals from the Soil Moisture Active Passive satellite (with a total of 69 anomalies). The locations of the anomalies are known. The amount of training samples varies between the signals but is mostly in the range of 1000.

**YahooS5:** The YahooS5 data collection contains four different data sets [2]. The A1 data set is based on real production traffic to some Yahoo properties. Data sets A2, A3 and A4 are synthetic data sets. In total, the four data sets contain 371 different signals and the locations of the anomalies are known. The amount of training samples also varies between the signals but is mostly in the range of 1000 as well. Some of the signals contain a large number of anomalies, which decreases the amount samples. We still found in our experiments that the model works quite well even on small numbers of training samples.

### 5.2 DATA PREPARATION

Before the training, we normalize the data and create training sequences by using a rolling window function, which creates sequences of length 100. These sequences are the inputs to the network. We move the window by one timestep, such that each point of the time series gets reconstructed 100 times at different places of the sequence. We then take the average of these reconstructions to obtain a single reconstruction point. Similarly, the *Critic* gives 100 different scores for each data point. We then take the median of those scores to obtain the *Critic* value at this points. (In the plots, we also show the 5th and 95th percentile of these 100 values to demonstrate the additional increase of variance in anomalous regions.)

In the case of the Yahoo data sets, we also apply a simple detrending function (which subracts the result of a linear least-squares fit to the signal) before training and testing, as many signals in these data sets contain linear trends.

Finally, we train the model only on non-anomalous sequences to ensure that the model does not learn to reconstruct anomalies, which is a well-known problem in reconstruction- and prediction-based anomaly detection methods.

### 5.3 ARCHITECTURE

In our experiments the inputs to our model are time series sequences of length 100 and the latent space is 20-dimensional.

Furthermore, we use a 1-layer bidirectional Long Short-Term Memory (LSTM) with 100 hidden units as an Encoder, and a 2-layer bidirectional LSTM with 64 hidden units each as the *Generator*. LSTMs, first introduced by Hochreiter & Schmidhuber (1997), are known for being able to model (long-term) time dependencies in data. Since we analyze time series data that naturally comes with these sequential dependencies, we chose LSTMs for our *Encoder* and *Generator*. We also use Dropout for the LSTM weights. We then use a 1-D convolutional net for both *Critics*, with the intention to capture local temporal features that can determine how anomalous a sequence is.

The model is trained on a specific data set for 2000 iterations, with a batch size of 64.

**Metrics**: We measure the performance of different methods with the commonly used metrics Precision, Recall and F1-Score. In our case, a true positive is found when we have overlap between

---

[1]The spacecraft telemetry data set can be downloaded here: `https://s3-us-west-2.amazonaws.com/telemanom/data.zip`

[2]The YahooS5 data collection can be requested here: `https://webscope.sandbox.yahoo.com/catalog.php?datatype=s&did=70`

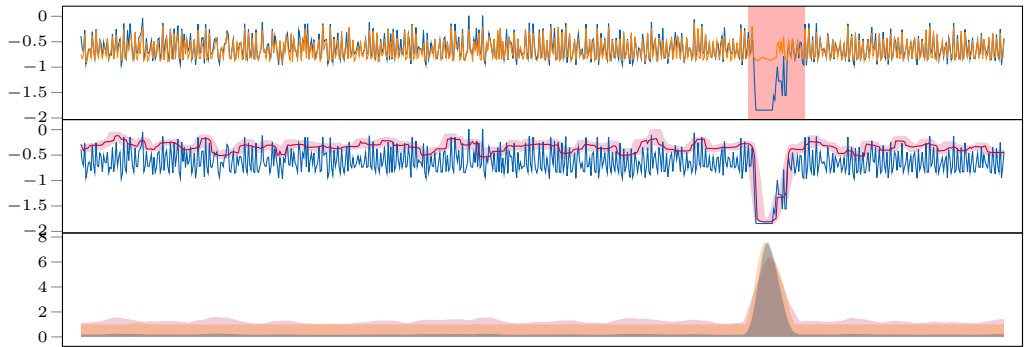

Figure 2: Results for the NASA S-1 signal. The first plot shows the original time series (blue) and the reconstructed one (orange), as well as the anomalous region (red). The second plot shows the time series and the aggregated output of the Critic (purple), as well as the 25th and 75th percentile of the Critic output. The third plot shows the reconstruction error (orange), the Critic error (purple) and the combined error (black)

a predicted anomalous sequence and a known sequence. A false positive is any sequence that was predicted but has no overlap with any known sequence, and a false negative is any known sequence that does not overlap with any of the predicted sequences.

## 5.4    RESULTS

The following section aims to evaluate different model variants in order to show the impact of our proposed model. To this end, we consider multiple variations of our system and check the scores on the data sets. This is going to show the differences between the combination of *Critic* and reconstruction errors compared to single measures alone. Further, we want to show how the proposed similarity measures can affect the performance of anomaly detection. After that, we compare our model to baseline methods in time series anomaly detection. (The full results can be seen in Tables 3 and 4 the appendix)

### 5.4.1    EVALUATION OF MODEL

As the results in Table 1 show, the *Critic* does indeed have an influence on the anomaly detection scores. On the NASA data sets, these differences are quite significant. For example, we see that in the SMAP data set, the best F1 score is achieved by using only the *Critic*.

When looking at the outputs of our different models, it becomes clear that the combination enforces the scores in anomalous regions while lowering the scores in normal regions. Figure 2 shows an example of this behavior. We also see this in the Precision and Recall differences, where we observe a greater average difference in the Precision than in the Recall when comparing a combination of both errors with the reconstruction error alone. This is due to the more significant reduction of false positives.

The differences are not as pronounced with the Yahoo data sets, which seems to be a result of the characteristics of those data sets, which are often more simple time series sequences with very clear, short and frequent anomalies. Although the scores are closer, we still observe that in many cases the combination of *Critic* and reconstruction error gives a better score compared to just the single measures. When looking at the visual outputs of our model, we also continue to see a generally more stable and reliable anomaly score in most cases, as the difference between the scores of anomalous regions compared to normal regions is increased.

In addition to that, our results show that in all but one data set, the similarity measures we proposed perform better than the point-wise reconstruction error. This shows how the use of local similarity

measures as a reconstruction error can be a more efficient approach than the point-wise error when dealing with time series.

| | **Data sets** | | | | | | | |
| | NASA | | | Yahoo S5 | | | | |
| **Variation** | MSL | SMAP | Total | A1 | A2 | A3 | A4 | Total |
| Critic | 0.292 | **0.706** | 0.599 | 0.037 | 0 | 0.01 | 0.014 | 0.013 |
| Critic + area difference | **0.573** | 0.689 | **0.655** | 0.524 | 0.789 | 0.945 | 0.873 | 0.862 |
| Area difference | 0.546 | 0.604 | 0.587 | 0.564 | 0.759 | **0.95** | **0.896** | **0.869** |
| Critic + point difference | 0.481 | 0.567 | 0.545 | 0.618 | 0.75 | 0.89 | 0.844 | 0.829 |
| Point difference | 0.466 | 0.456 | 0.462 | **0.62** | 0.747 | 0.867 | 0.843 | 0.817 |
| Critic + DTW | 0.503 | 0.596 | 0.573 | 0.612 | **0.798** | 0.897 | 0.826 | 0.831 |
| DTW | 0.434 | 0.494 | 0.481 | 0.603 | 0.777 | 0.876 | 0.832 | 0.82 |

Table 1: F1-Scores of different variations of our model.

### 5.4.2 COMPARISON TO BASELINES

We now compare our model to a simple LSTM-based prediction, an ARIMA-based prediction model and a simple Autoencoder with dense layers (in order to represent a network without LSTMs). A short description of the baseline methods can be found in appendix A.2.

As is evident in Table 2, our proposed GAN models perform better on three of the six individual data sets and, when considering all signals across the NASA and YahooS5 data sets, outperform every other method. For the three datasets where they perform better, we see a significant 12% improvement on SMAP, a 15% improvement in A3 and a 27% improvement in A4.

| | **Data sets** | | | | | | | |
| | NASA | | | Yahoo S5 | | | | |
| **Method** | MSL | SMAP | Total | A1 | A2 | A3 | A4 | Total |
| LSTM prediction | **0.602** | 0.521 | 0.547 | **0.706** | 0.783 | 0.822 | 0.706 | 0.764 |
| Arima prediction | 0.355 | 0.452 | 0.416 | 0.678 | **0.898** | 0.498 | 0.621 | 0.621 |
| Dense Autoencoder | 0.517 | 0.629 | 0.593 | 0.599 | 0.206 | 0.297 | 0.26 | 0.29 |

Table 2: F1-Scores of baseline models

## 6 CONCLUSION

In this paper we showed how GANs can be effectively used for anomaly detection in time series data. We proposed a cycle-consistent GAN architecture that allows the encoding and decoding of time series. We further proposed new reconstruction error measures based on local similarities that outperform the point-wise reconstruction. We have also shown that a combination of the *Critic* output and the reconstruction error can help to reduce the number of false positives, as well as might increase the number of true positives. This is due to the more robust anomaly scores that our method is providing, where we effectively increase the distance between the scores of anomalous and non-anomalous regions. As a result, the anomaly scores are more robust and allow a better detection of anomalous time series sequences.

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

# A APPENDIX

## A.1 RESULT DETAILS

| Variation | | NASA | | | Yahoo S5 | | | | |
|---|---|---|---|---|---|---|---|---|---|
| | | MSL | SMAP | Total | A1 | A2 | A3 | A4 | Total |
| Critic | P | 0.869 | 0.8 | 0.806 | 0.1 | 0 | 0.143 | 0.192 | 0.11 |
| | R | 0.176 | 0.633 | 0.476 | 0.022 | 0 | 0.005 | 0.007 | 0.007 |
| | F1 | 0.292 | 0.706 | 0.599 | 0.037 | 0 | 0.01 | 0.014 | 0.013 |
| Critic + area difference | P | 0.674 | 0.598 | 0.612 | 0.557 | 0.689 | 0.969 | 0.936 | 0.885 |
| | R | 0.5 | 0.816 | 0.705 | 0.494 | 0.924 | 0.921 | 0.818 | 0.84 |
| | F1 | 0.573 | 0.689 | 0.655 | 0.524 | 0.789 | 0.945 | 0.873 | 0.862 |
| Area difference | P | 0.603 | 0.495 | 0.518 | 0.535 | 0.638 | 0.968 | 0.949 | 0.871 |
| | R | 0.5 | 0.773 | 0.676 | 0.596 | 0.937 | 0.932 | 0.849 | 0.867 |
| | F1 | 0.546 | 0.604 | 0.587 | 0.564 | 0.759 | 0.95 | 0.896 | 0.869 |
| Critic + point difference | P | 0.69 | 0.478 | 0.513 | 0.623 | 0.628 | 0.967 | 0.929 | 0.869 |
| | R | 0.37 | 0.696 | 0.581 | 0.612 | 0.93 | 0.824 | 0.773 | 0.793 |
| | F1 | 0.481 | 0.567 | 0.545 | 0.618 | 0.75 | 0.89 | 0.844 | 0.829 |
| Point difference | P | 0.707 | 0.357 | 0.401 | 0.596 | 0.618 | 0.97 | 0.938 | 0.863 |
| | R | 0.352 | 0.633 | 0.543 | 0.646 | 0.943 | 0.784 | 0.765 | 0.777 |
| | F1 | 0.466 | 0.456 | 0.462 | 0.62 | 0.747 | 0.867 | 0.843 | 0.817 |
| Critic + DTW | P | 0.752 | 0.514 | 0.548 | 0.636 | 0.702 | 0.971 | 0.936 | 0.889 |
| | R | 0.38 | 0.71 | 0.6 | 0.59 | 0.924 | 0.833 | 0.739 | 0.781 |
| | F1 | 0.503 | 0.596 | 0.573 | 0.612 | 0.798 | 0.897 | 0.826 | 0.831 |
| DTW | P | 0.665 | 0.402 | 0.438 | 0.6 | 0.664 | 0.978 | 0.945 | 0.879 |
| | R | 0.324 | 0.643 | 0.533 | 0.607 | 0.937 | 0.794 | 0.743 | 0.769 |
| | F1 | 0.434 | 0.494 | 0.481 | 0.603 | 0.777 | 0.876 | 0.832 | 0.82 |

Table 3: Precision, Recall and F1-Scores of different variations of our model.

| Method | | NASA | | | Yahoo S5 | | | | |
|---|---|---|---|---|---|---|---|---|---|
| | | MSL | SMAP | Total | A1 | A2 | A3 | A4 | Total |
| LSTM prediction | P | 0.463 | 0.38 | 0.406 | 0.614 | 0.643 | 0.93 | 0.837 | 0.814 |
| | R | 0.861 | 0.826 | 0.838 | 0.831 | 1 | 0.737 | 0.611 | 0.72 |
| | F1 | 0.602 | 0.521 | 0.547 | 0.706 | 0.783 | 0.822 | 0.706 | 0.764 |
| Arima prediction | P | 0.268 | 0.359 | 0.324 | 0.614 | 0.912 | 0.908 | 0.892 | 0.845 |
| | R | 0.528 | 0.609 | 0.581 | 0.758 | 0.885 | 0.343 | 0.476 | 0.491 |
| | F1 | 0.355 | 0.452 | 0.416 | 0.678 | 0.898 | 0.498 | 0.621 | 0.621 |
| Dense Autoencoder | P | 0.682 | 0.709 | 0.701 | 0.837 | 1 | 0.988 | 0.687 | 0.836 |
| | R | 0.417 | 0.565 | 0.514 | 0.466 | 0.115 | 0.175 | 0.16 | 0.176 |
| | F1 | 0.517 | 0.629 | 0.593 | 0.599 | 0.206 | 0.297 | 0.26 | 0.29 |

Table 4: Precision, Recall and F1-Scores of baseline models

## A.2 BASELINE METHODS

**LSTM prediction:** The network consists of two LSTM layers with 80 units each and a following Dense layer with one unit which predicts the next time step. We then use a point-wise prediction error between the predicted and the observed time step to identify outliers. The model is similar to the one used by Hundman et al. (2018).

**ARIMA:** ARIMA models try to describe autocorrelations in the time series, i.e. how the values of past time steps influence the current and future time steps. We use an ARIMA model to predict the next time steps based on the previous ones. The parameters were selected individually for each data set based on the best performance. We then use a point-wise prediction error to detect outliers.

**Autoencoder:** The Autoencoder model consists of three Dense layers with 60, 20 and 60 units respectively. Again, a point-wise reconstruction error is used in order to detect anomalies. The method is related to the one used by Malhotra et al. (2016), however we are not using LSTMs in this case since our GAN model with only the reconstruction error is essentially an LSTM-based Autoencoder and this way we include a baseline with a different layer type as well.

### A.3 EXAMPLES

The following examples show how our method performs on several signals. The plots are structured the following: The first plot shows the original time series (blue) and the reconstructed time series (orange), as well as the anomalous region (red). The second plot shows the time series and the aggregated output of the *Critic* (purple), as well as the 25th and 75th percentile of the Critic output. The third plot shows the reconstruction error (orange), the *Critic* error (purple) and the combined error (black)

#### A.3.1 NASA SMAP

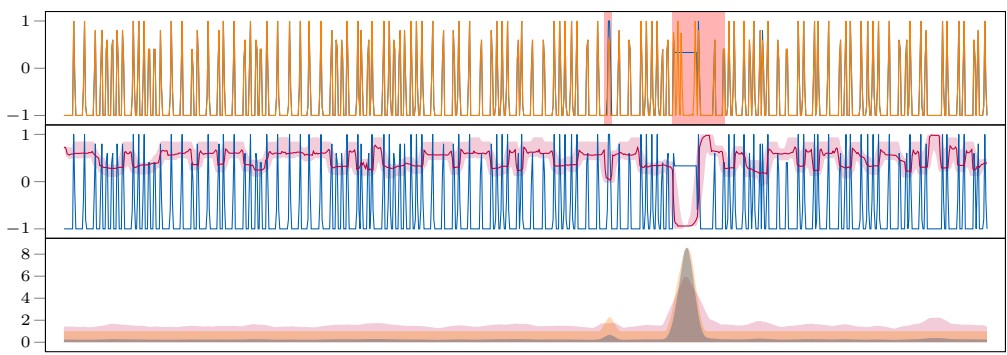

Figure 3: E-1 Signal

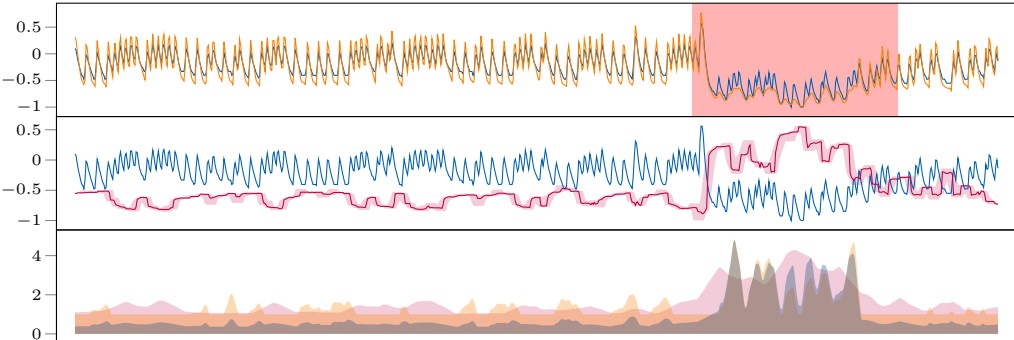

Figure 4: E-9 Signal

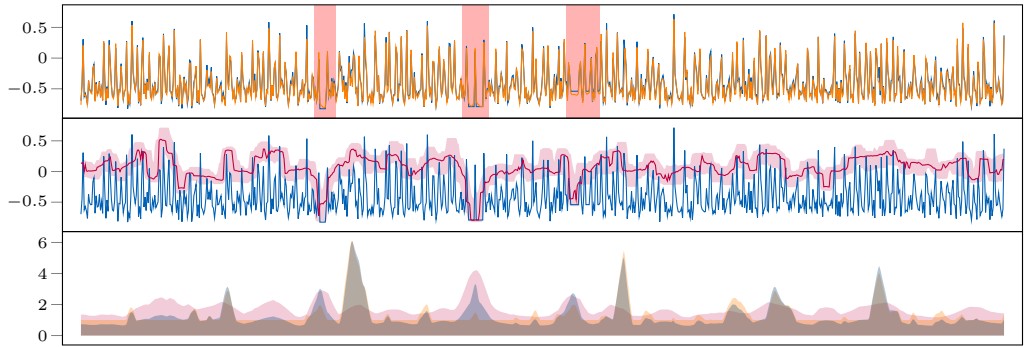

Figure 5: P-1 Signal

## A.3.2 NASA MSL

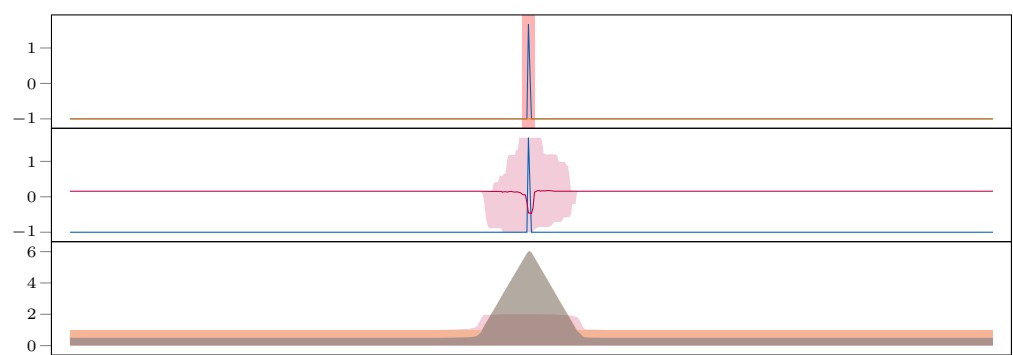

Figure 6: S-2 Signal

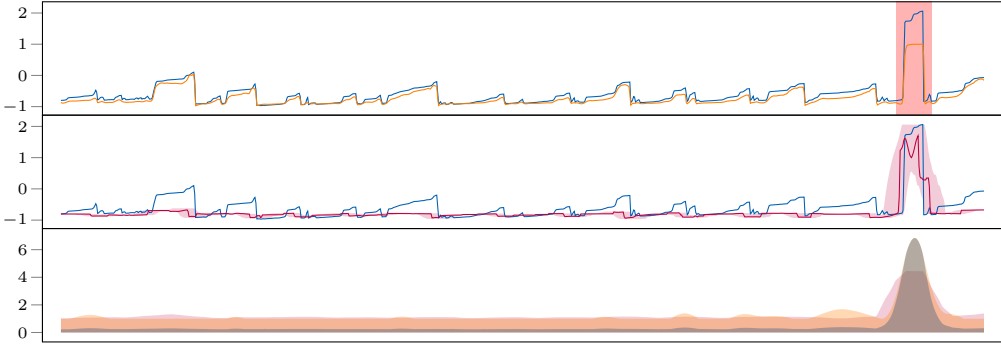

Figure 7: F-5 Signal

### A.3.3   YAHOO A1

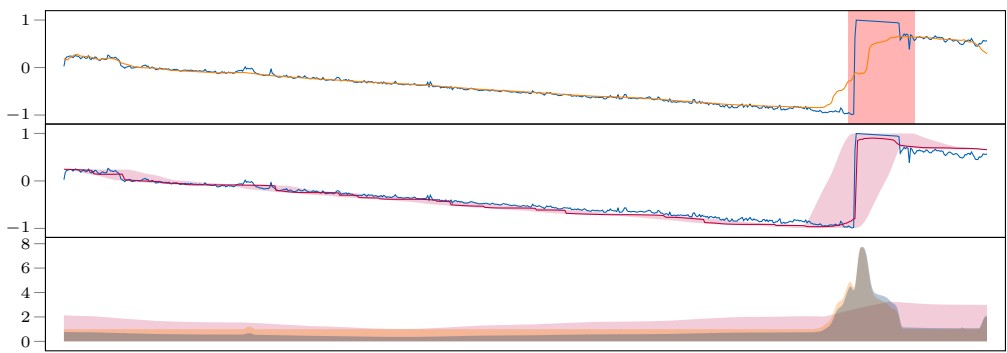

Figure 8: real-32 Signal

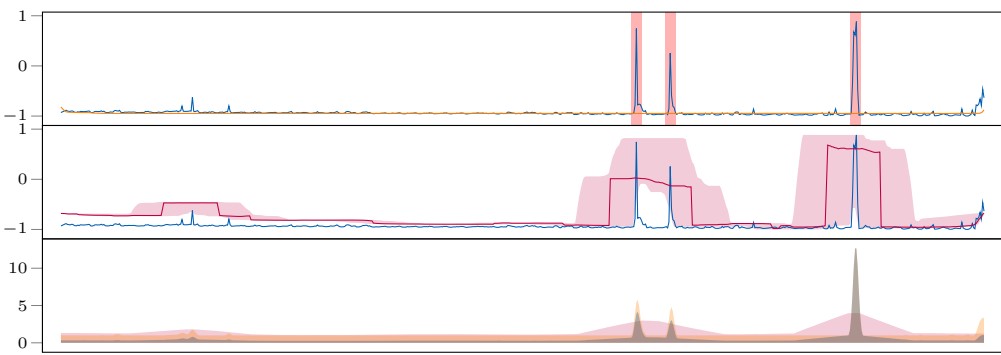

Figure 9: real-8 Signal

### A.3.4 YAHOO A2

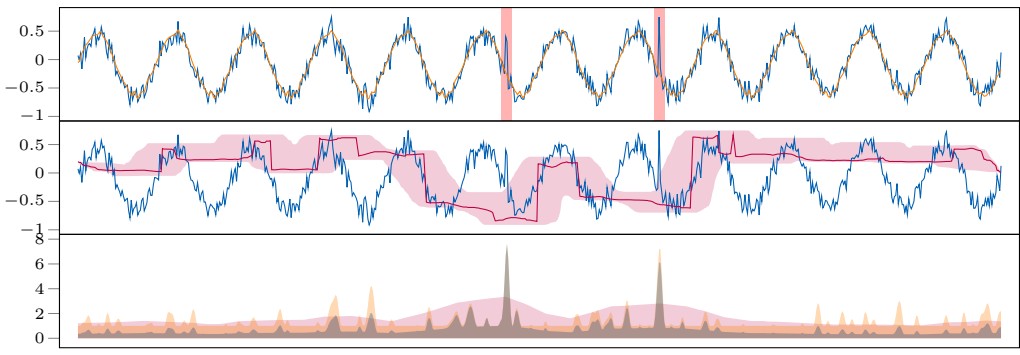

Figure 10: synth-40 Signal

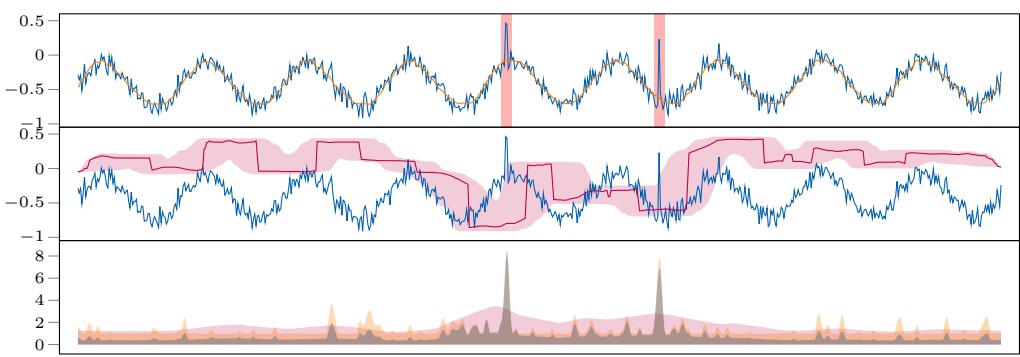

Figure 11: synth-52 Signal

### A.3.5 YAHOO A3

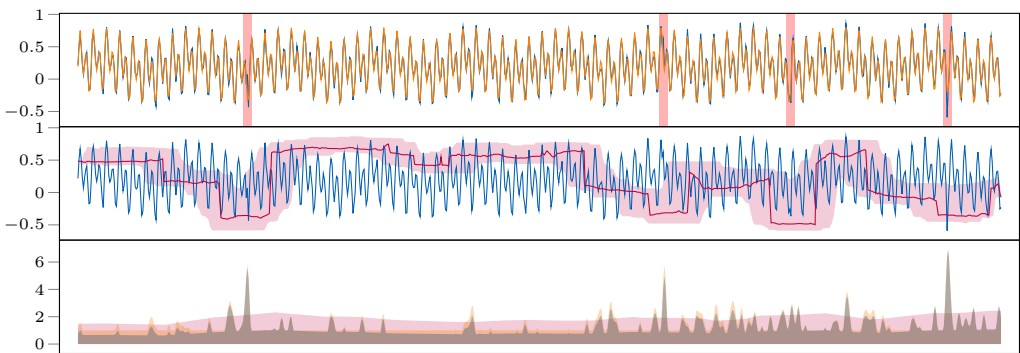

Figure 12: TS-29 Signal

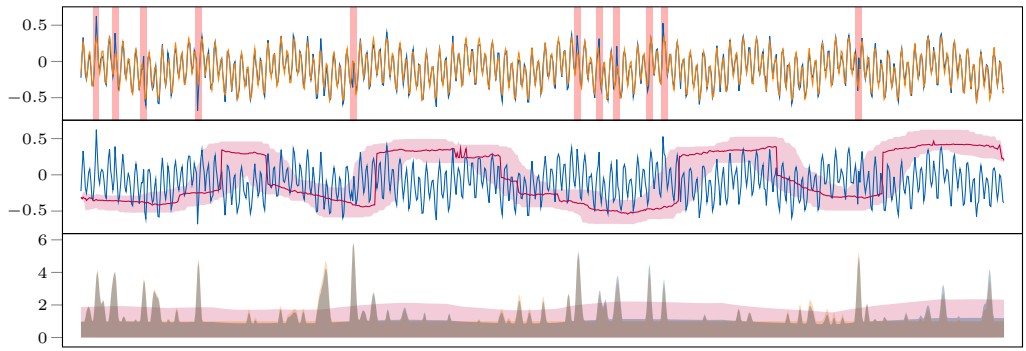

Figure 13: TS-1 Signal

### A.3.6 YAHOO A4

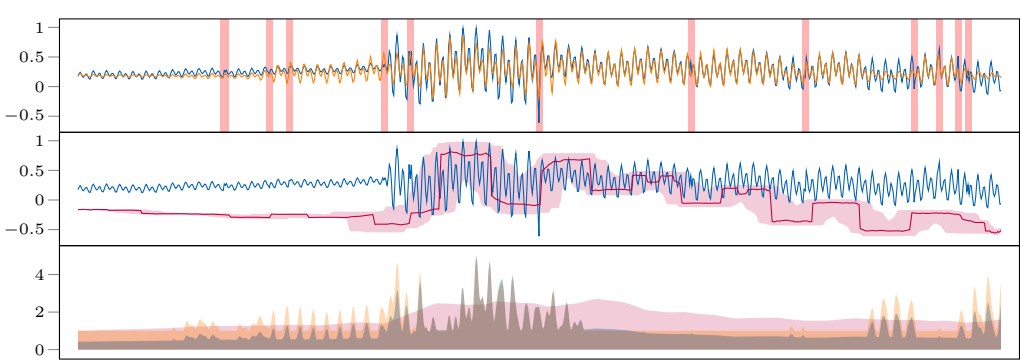

Figure 14: TS-8 Signal

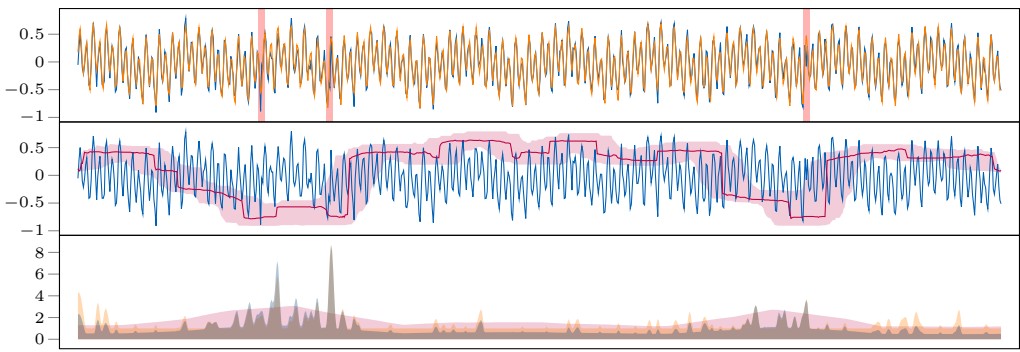

Figure 15: TS-61 Signal

