# OpenReview forum: "Adversarially learned anomaly detection for time series data"
_ICLR.cc/2020/Conference — Reject_

### Official Review · AnonReviewer1 · 2019-10-19
**Official Blind Review #1**

**Rating:** 1

**Review:**

The paper trains a GAN on univariate time series data and uses reconstruction errors in combination with the critic's output to predict anomalous subsequences. The method is applied on two real-world data sets and compared to three simple baselines.

I have several reservations about this manuscript:
- The methodology isn't very original: The GAN architecture is essentially a slightly modified CycleGAN trained with Wasserstein loss.
- Many design choices appear to be ad-hoc: there has been no principled selection of the GAN's hyperparameters, nor has their effect on the experimental results been studied. For computing the reconstruction error the authors use the integral over the difference between two time series without taking the absolute value, which is a very unusual choice. (Standard choices would be e.g. L1 or L2 norm.) While this turns out to have worked "surprisingly well" on the studied datasets, it is not difficult to construct scenarios where this choice will fail.
- Another design choice that should be discussed in more detail is the smoothing of the time series data using a moving average filter; in the context of anomaly detection this can have a significant impact, and it may not work equally well for different data sets, so a principled approach for determining the level of filtering is paramount. Same goes for the de-trending, and the actual parameters both of the moving averages and the de-trending functions should be reported.
- It is not clear to me how exactly the anomaly scores (supposedly for different sequence lengths l?) are used to predict the subsequence(s) containing an anomaly. It seems there is no incentive to keep the predicted subsequences as short as possible, i.e. if the anomaly score indicates there might be an anomaly present, then the safe approach is to just flag the entire sequence as anomalous (it doesn't hurt the sensitivity they way it's computed)?
- The description of the experiments requires more detail. Are both datasets labelled? After dividing the datasets into rolling chunks of length 100, how many samples do the training and test sets contain? How many of the test set samples contain anomalies?
- The baselines need to be described in more detail. What method was used, e.g. to select and fit the ARIMA models? What type of reconstruction loss was used? How were the anomalous subsequences predicted?
- Is there any way to compare the proposed method with Li et al's (who also used a GAN for anomaly detection in time series data)?

Detailed comments:
- abstract: "particular hard" -> "particularly hard"
- p.3, paragraph starting with "To support...": shouldn't this be "E(x) with x ~ P_X"?
- p.4 and p.6: broken citations "?"

------------------

I acknowledge that I have read the authors' response, but it doesn't change my assessment that several major revisions are needed:
- further motivating the design choices and comparing them to the state-of-the-art;
- formalizing (ideally providing a model for) the sort of anomalies that the proposed method aims to detect;
- discussing limitations of the proposed approach.

**Experience Assessment:**

I have published one or two papers in this area.

**Review Assessment: Checking Correctness Of Derivations And Theory:**

N/A

**Review Assessment: Checking Correctness Of Experiments:**

I assessed the sensibility of the experiments.

**Review Assessment: Thoroughness In Paper Reading:**

I read the paper at least twice and used my best judgement in assessing the paper.

---

> ### Author Response · Authors · 2019-11-14
> **Response to Reviewer #1**
>
> Thank you very much for your detailed comments. We updated the draft and included many of the mentioned points.
>
> In particular, we want to respond to some of the points that were mentioned:
> - While the GAN architecture itself is not very novel, we want to emphasize that this is not our main contribution and we never claimed that it is a novel approach. However, we have adopted the architecture and applied it to time series signals, which is a rather novel method as far as we know. Furthermore, we introduce two similarity measures instead of a point-wise reconstruction error and show it can improve the scores and we demonstrate in much detail how a combination of the Critic and the reconstruction error can give much more robust anomaly detection scores.
> - We have added some comments to the paper that try to clarify some of the design choices. And while the used similarity measures are unusual, we want to emphasise that this is exactly the claim of the paper, as we not only introduced those in the time series anomaly detection segment, but also show how these measures can improve the anomaly detection scores (see section 5.4.1 for the in-depth analysis). We are also not aware of any scenarios where this choice will give much worse scores than a standard L1 or L2 norm, therefore we would be happy if these pitfalls would be pointed out to us. We rather think that the experiments are actually a good indication that the proposed measures work very well in practice.
> - We have rewritten parts of the experiments section such that it becomes clearer what the individual steps are. We also added a short description of the baseline methods in the appendix.
> - We are not completely sure what is meant by ‘flagging the entire sequence as anomalous’ is the safe approach. Once we computed the anomaly scores for each time step, we try to identify any outliers by statistical analysis. If we find outliers, we flag them as an anomaly based on the location. We are currently not sure what benefit it would have to flag the entire sequence. We have added some more explanations to the corresponding section of the paper and hope that this clarifies any open points.
> - We compare our method conceptually with Li et al’s approach in the related work section. Since we modified the structure to a CycleGAN, but the proposed ideas are similar when it comes to the use of the Critic and the point-wise reconstruction error, we did not include this method in our benchmarks since our evaluation in section 5.4.1 already includes these features.
>
> Thank you for pointing out the additional typos and citation errors, we fixed them in the revised draft.

---

### Official Review · AnonReviewer2 · 2019-10-21
**Official Blind Review #2**

**Rating:** 3

**Review:**

This paper proposes a GAN model with cycle-consistent loss function for anomaly detection on timeseries. The loss function is the combination of reconstruction error (L2 norm) and both discriminators (generator and decoder) loss. Once trained, the anomaly score is computed as the mean-var-normalized product of the reconstruction score and the discriminator score. The framework is adapted to timeseries by using LSTM networks for the generator and the encoder and 1D CNN for the discriminators (critics). For the calculation of the reconstruction error, the authors study three approaches: point-wise, dynamic time warping and area difference. The method is applied to two datasets (NASA Spacecraft telemetry, Yahoo traffic) and is compared to simple baselines (LSTM, ARIMA, DenseAE). The results shows an improvement over the baselines. Variations of the models are also studied (with and without critic, different similarity measures for the reconstruction errors). The results show that the discrimination score (critic) does overall provide an improvement over the reconstruction error alone. And for the reconstruction score, the area differences is best, followed by DTW and point-wise.

PROS:

* A somewhat novel approach to anomaly detection in timeseries is proposed that combines GAN discriminator score with a simple area-based similarity measure for the  reconstruction error.
* A solid adaptation of CycleGAN to timeseries.
* The simple area-based similarity measure is interesting and seems to be a good match for timeseries reconstruction as slight shifts are not penalized.
* The paper is well written, well organized and easy to follow.
* The technical content is sound and the math correct.


CONS:

* not really novel (CycleGAN for timeseries) nor DTW as reconstruction error.
* no comparison to State-of-the-art GAN models for anomaly detection, such as AnoGAN and ADGAN.
* Too few datasets are used (2) for the experiments, making it hard to draw conclusions as to which variation is better.
* Baseline approaches are just mentioned by name. A minimal description would be desirable.
* The ground-truth anomalies and the anomaly threshold are not marked in the plots, making it hard to evaluate them.

Overall the paper proposes a method that improves over baseline but is not compared to other GAN-based SOT models. The novelty is not very high as the main architecture is from CycleGAN and the proposed similarity measures for the reconstruction error are not really novel.



**Experience Assessment:**

I have published one or two papers in this area.

**Review Assessment: Checking Correctness Of Derivations And Theory:**

I assessed the sensibility of the derivations and theory.

**Review Assessment: Checking Correctness Of Experiments:**

I carefully checked the experiments.

**Review Assessment: Thoroughness In Paper Reading:**

I read the paper thoroughly.

---

> ### Author Response · Authors · 2019-11-14
> **Response to Reviewer #2**
>
> Thank you very much for the in-depth response. We have updated our draft and included many of the mentioned points.
> Specifically, we have added a small description of the baseline methods and marked the ground truth anomalies in the plots.
>
> We also want to address some specific issues that were pointed out:
>
> At first we would like to clarify some things about the novelty of our method. We think our approach is slightly more novel than mentioned by the reviewer, since it’s the first use of a CycleGAN in the time series domain as well it’s the first use of other similarity measures as a reconstruction error. We are not aware of any work that uses DTW in the sense of a reconstruction error. Also, there is no other work known to us that investigates the performance of these different reconstruction errors in this depth (see section 5.4.1 where we discuss the different reconstruction methods in detail and how our proposed similarity measures outperform the point-wise error in the experiments).
>
> Furthermore, we did not include AnoGAN and ADGAN in our experiments as they are GAN based anomaly detection methods for a different domain and not necessarily time series. Therefore, we consider our model as an adaption of these methods to the time series domain and only compare our model to other time series anomaly detection methods.
>
> We also think that the experiments of the paper are enough to draw conclusions since they provide a good comparison between the methods on a large number of different signals and it becomes clear that the proposed similarity measures as well as our combination with the Critic are performing very well. As we already mentioned in our response to reviewer #3, we want to emphasise that labeled time series anomaly detection data sets are rare to find and we want to point out that many of the related work also uses only a very limited number of data sets (e.g. Li et al., Zhou et al. (2019) and Hundman et al. (2018)).

---

### Official Review · AnonReviewer3 · 2019-10-22
**Official Blind Review #3**

**Rating:** 1

**Review:**

Summary:  The paper propose a cycle gan variants combined with RNN for time series anomaly detection. The setting is assuming training model on a given normal data, then applying the trained model to detect anomalies.  Different detecting criteria are studied in the experiments.

1. It is not clear the advantage of using GANs in anomaly detection of the proposed algorithm, and it is questionable if using GANs is really useful. The authors only provide hand-waving explanation.  For example, the papers says  " Therefore, once the Critic is trained, it should assign more or less stable scores to the normal sequences and a significantly different score to an anomalous sequence." in p5.  I'm doubt if it is true.  The GAN objective only says the score on x~p(x) and g(z) are similar. Also, the score for normal samples can also have a distribution since we only do mean matching in W-1 distance.  I think a stronger analysis with certain assumptions is necessary for making this statement and justifying the proposed algorithm.  Some possible route can be found in

Chang et al., Kernel Change-Point Detection with Auxiliary Deep Generative Models, ICLR 2019.

Although their setting is slightly different, where they focus on change point detection.  They provide a testing power lower bound explanation of using the critic of GANs, but they require some early stopping. Otherwise the guarantee won't hold.  I'm wondering if the analysis can be extended to here. Also, if the proposed algorithm requires the early stopping or not? If not, why?

2. The experiments are not conclusive.  The simple LSTM predictions beats the proposed algorithms in some cases.   Given there are only two datasets, it's hard to say if the proposed algorithm is really better. Also, the author proposed so many combinations, it is also not clear which one should be favored based on the table.

3. Many related works are missing.  In addition to Chang el al (2019) mentioned above, there are many related works of using GANs in time series detection problem, e.g. BeatGAN: Anomalous Rhythm Detection using Adversarially Generated Time Series, and many others.


To summarize, the motivation and the advantage of using GANs is not well justified in addition to experiments.  Also, the authors only study two datasets, and the results are not very conclusive.

**Experience Assessment:**

I have published one or two papers in this area.

**Review Assessment: Checking Correctness Of Derivations And Theory:**

I carefully checked the derivations and theory.

**Review Assessment: Checking Correctness Of Experiments:**

I carefully checked the experiments.

**Review Assessment: Thoroughness In Paper Reading:**

I read the paper thoroughly.

---

> ### Author Response · Authors · 2019-11-14
> **Response to Reviewer #3**
>
> Thank you very much for the thorough feedback. We will try to address the mentioned points from our point of view and hope that this clarifies some issues.
>
> 1. While it is true that the paper is not going into the details when it comes to the Critic output and the justification behind using it as an anomaly score, we see in the experiments that the output of the Critic can practically be used as an anomaly score, since the anomalous regions receive different scores than the normal regions. This is based on the same intuition that was already used in some of the cited previous papers which use GANs as an anomaly measure (e.g. Li et al (2018)). Basically, the Critic is trained to separate the generated time series sequences from the real ones, meaning it learns to assign scores of how real a time series sequence is. Using this trained Critic, we can detect anomalies since it will assign different scores to those sequences compared to the normal ones as shown in related literature and in our experiments.
> Regarding the work of Chang et al. (2019), we are not sure at the moment how much it actually relates to our work since they train an ‘auxiliary generative model’ as emphasised in their appendix, which has some different constraints. We will look into this paper in more detail and try to identify any overlaps with our work.
> Lastly, we do not use early stopping in our current implementation and we are not sure if this would help in our case since the model is currently already performing as expected and provides quite convincing scores as shown in the experiments.
>
> 2. It is also true that the number of data sets is rather limited in this paper. However, in our opinion the experiments already show very convincing results by reaching better scores in the majority of our experiments.  It is also the case that labeled time series anomaly detection data sets are rare to find and we want to point out that many of the related work also use only a very limited number of data sets (e.g. Li et al., Zhou et al. (2019) and Hundman et al. (2018)).
> In addition, it is also confirmed in the plots how our method is improving the anomaly scores by increasing the distance between the scores of anomalies and non-anomalies, leading to much more robust anomaly scores.
> Regarding the large number of combinations, we wanted to give a comprehensive overview of how the different similarity measures perform on the data sets. The main result of this is that the point-wise error, which is used in many previous papers (e.g. Hundman et al., 2018; Malhotra et al., 2015, Li et at., 2018), might not be the best option in most cases and we show how two different similarity measures can improve the scores (see section 5.4.1 where we discuss the different reconstruction methods in detail and how our proposed similarity measures outperform the point-wise error in the experiments).
>
> 3. Thank you for pointing it out, we have improved the related literature section accordingly.
>
> In summary, we think that our approach is justified through previous work and the results of the experiments. Even though there are not more data sets used in this work, we think that the results are already conclusive in the sense that they provide solid results and a comprehensible evaluation of different methods. They show how similarity measures can be used in time series anomaly detection to give much better reconstruction errors, as well as how they can be combined with a Critic to give robust overall anomaly scores.

---

### Decision · Program_Chairs · 2019-12-19

**Decision:**

Reject

**Comment:**

The paper proposes a cycle-consistent GAN architecture with measuring the reconstruction error of time series for anomaly detection.

The paper aims to address an important problem, but the current version is not ready for publication. We suggest the authors consider the following aspects for improving the paper:
1. The novelty of the proposed model: motivate the design choices and compare them with state-of-art methods
2. Evaluation: formalize the target anomalies and identify datasets/examples where the proposed model can significantly outperform existing solutions.